environmental chemistry/nanotechnology/
inorganic chemistry

pollution, heavy metals, environmental
remediation, inorganic nanomaterials, adsorption

**Author for correspondence:**
Michael B. Mensah
e-mails: michael.mensah@knust.edu.gh,
mike_baa@yahoo.com

This article has been edited by the Royal Society of Chemistry, including the commissioning, peer review process and editorial aspects up to the point of acceptance.

# Heavy metal pollution and the role of inorganic nanomaterials in environmental remediation

Michael B. Mensah[1], David J. Lewis[2],
Nathaniel O. Boadi[1] and Johannes A. M. Awudza[1]

[1]Department of Chemistry, Kwame Nkrumah University of Science and Technology, PMB, Kumasi, Ghana [2]Department of Materials, University of Manchester, Oxford Road, M13 9PL, UK

MBM, 0000-0002-8080-4812

Contamination of water and soil with toxic heavy metals is a major threat to human health. Although extensive work has been performed on reporting heavy metal pollutions globally, there are limited review articles on addressing this pernicious phenomenon. This paper reviews inorganic nanoparticles and provides a framework for their qualities required as good nanoadsorbents for efficient removal of heavy metals from water. Different inorganic nanoparticles including metals, metal oxides and metal sulfides nanoparticles have been applied as nanoadsorbents to successfully treat water with high contaminations of heavy metals at concentrations greater than $100 \text{ mg l}^{-1}$, achieving high adsorption capacities up to $3449 \text{ mg g}^{-1}$. It has been identified that the synthesis method, selectivity, stability, regeneration and reusability, and adsorbent separation from solution are critical parameters in deciding on the quality of inorganic nanoadsorbents. Surface functionalized nanoadsorbents were found to possess high selectivity and capacity for heavy metals removal from water even at a very low adsorbent dosage of less than $2 \text{ g l}^{-1}$, which makes them better than conventional adsorbents in environmental remediation.

## 1. Introduction

Pollution of water and soil is a global concern. Heavy metals such as mercury (Hg), lead (Pb), cadmium (Cd) and arsenic (As) are highly poisonous and their pollution even at low concentrations in water and soil poses a serious threat to the health of humans and other terrestrial animals. Mercury has deleterious effects on the human kidney, brain and central nervous system, whereas

lead pollution has been associated with neurodevelopmental effects, and cadmium is classified as a carcinogen [1,2]. Arsenic intake causes dermal lesions, skin cancers, bladder and lung cancers [1]. To mitigate these health effects, the World Health Organization (WHO) has set recommended levels for Hg, Pb, Cd and As in drinking water to be 0.006, 0.01, 0.003 and 0.01 mg $l^{-1}$, respectively [1]. The reported major source of mercury contamination in most developing countries is the proliferation of illegal artisanal gold mining activities in which mercury-gold amalgamation methods are used [3–5]. Manufacturing of batteries, steel, plastics and fertilizers, and mining are the common sources of the exposure of lead and cadmium to the environment. Arsenic contamination is predominant in underground waters where sulfide and sedimentary deposits are present [1].

Mercury is volatile and, predominantly, exists as $Hg^{2+}$ in water [6,7]. The $Hg^{2+}$ ion is highly toxic even at low concentrations and can accumulate in ecosystems, in particular apex predators, causing several disorders and diseases. Methyl mercury ($CH_3Hg^+$) is another form of mercury that bioaccumulates in organic tissues, poisoning the food chain [8,9]. The Minamata Convention on Mercury established in October 2013 saw over 130 countries signing, and declared mercury pollution as a global problem that no country can solve alone is a classic example of food poisoning by heavy metals [10]. This convention was inspired by a fishing village near Minamata city in Japan, which drew the world's attention to the devastating effects of mercury as a powerful neurotoxicant that is known to dangerously affect fetuses, infants and young children. A local chemical factory was eventually found responsible for discharging mercury waste into the Shiranui Sea which killed fishes and other sea creatures and mysteriously, affected several families that ate contaminated seafoods. Children were diagnosed with cerebral palsy and eventually with what was known as the Minamata disease. The disaster is outlined by Kessler [10].

Inorganic lead ($Pb^{2+}$) compounds are the most predominant forms of lead in the environment. The highly toxic organic lead compounds, associated with the historic use of leaded gasoline and paints have been phased out [11]. Cadmium ions ($Cd^{2+}$), primarily, exists as greenockite (hexagonal cadmium sulfide, CdS) and usually occurs in association with zinc [12]. Arsenic is associated with gold ores and exists in different oxidation states (−3, 0, +3 and +5) and in water, the $As^{5+}$ form (arsenate) is predominant [1].

Significant research has been reported widely on the levels of contamination of heavy metals in the environment, and these lay the foundation for developing sequestration strategies for remediation. For example, Ghana being an African country where mining and quarrying activities are predominant, the levels of Hg, Pb, Cd and As pollution in water bodies, sediments and soils reported from 2003 to 2018, ranged from 0.001 µg $l^{-1}$–19.82 mg $l^{-1}$, 0.001–93.10 mg $kg^{-1}$ and 0.0011–57.80 mg $kg^{-1}$ of Hg, 0.012 µg $l^{-1}$–11.60 mg $l^{-1}$, 0.005–307.20 mg $kg^{-1}$ and 1.025–571.3 mg $kg^{-1}$ of Pb, 0.006 µg $l^{-1}$–1.4 mg $l^{-1}$, 0.00015–90.50 mg $kg^{-1}$ and 0.0011–103.66 mg $kg^{-1}$ of Cd and 0.011 ug $l^{-1}$–18.40 mg $l^{-1}$, 0.00197–10 200 mg $kg^{-1}$ and 0.08–18.60 mg $kg^{-1}$ of As, respectively [13–70]. The source of pollution has been largely attributed to illegal artisanal gold mining activities.

Heavy metal removal strategies from water include chemical precipitation, ion exchange, solvent extraction, cloud point extraction, ultrafiltration, reverse osmosis, crystallization, evaporation, photo-catalysis and adsorption techniques. Apart from adsorption, the rest of the techniques suffer disadvantages such as (i) sludge production, (ii) high energy requirements, (iii) high operational costs, (iv) requirement for the use of solvents, (v) possible inefficiency at removing trace concentrations of some toxic metal ions, (vi) solubility limitations, (vii) possible use of high-pressure operations, and (viii) requirement of expensive analytical instrumentation [8,71–74]. However, the adsorption technique is simple and effective; the adsorbents are usually easy to handle, and the operation and design are flexible [8,75].

The most common materials used as adsorbents for heavy metals sequestration are activated carbon, zeolite, silica gel, activated alumina, natural clay (kaolinite, bentonite, illite, stevensite and rectorite) and biomass [9,73]. Silica has been shown to possess a surface area of 200–1500 $m^2 g^{-1}$ and, together with its porous nature, appears to be a very good adsorbent for heavy metal remediation [9]. Activated carbon with a surface area of 2082 $m^2 g^{-1}$ has been shown to display the maximum adsorption capacity of 56.2–135.8 mg $g^{-1}$ over other equally good adsorbent systems [76]. Although these adsorbent materials are suitable, they lack selectivity and have a poor affinity towards targeted heavy metals. With the advent of nanoscience, which involves techniques for scaling down sizes of materials to the nanoscale (1–100 nm), adsorbents can be designed to possess enhanced properties compared with common bulk material adsorbents.

Nanoscale adsorbents offer better selectivity, capacity and improved affinity towards heavy metals pollutants and also provide faster and efficient adsorption processes [77]. In addition, nanoscale

adsorbents possess (i) high surface area to volume ratio, (ii) surface functional groups tailored for a specific application, (iii) short diffusion route or absence of internal diffusion resistance, (iv) high porosity, (v) enhanced structural properties, and (vi) catalytic properties [8,72,73,77–79]. These properties make nanoadsorbents highly selective, and with the capacity to achieve efficient adsorption of a variety of water pollutants including heavy metals. Examples of useful nanoadsorbents are zero-valent iron (n-ZVI), iron oxide ($Fe_3O_4$), $Fe_3O_4@SiO_2$-SH, iron sulfide (FeS), amorphous silica ($SiO_2$) and aluminium-silicate-mixed oxides nanomaterials. Interestingly, blends of inorganic nanomaterials with large surface area metal-organic frameworks (MOFs) and covalent organic frameworks (COFs) have shown high adsorption capacity for heavy metals and radionuclides [80,81].

This review covers several key areas including the different types of inorganic nanoparticles useful as adsorbent materials and the potential qualities of nanoparticles required for efficient removal of heavy metals from solution. This knowledge is essential for the design of efficient nanoadsorbents with capacities to remove heavy metals at levels prevailing in the environment and subsequently providing improvement to water treatment strategies, especially in mining areas.

# 2. Inorganic nanomaterials used as adsorbents

Nanomaterials are nanoscale materials with improved performance relative to the bulk material, often called emergent properties. Nanoscale materials due to the size effect emanating from the quantum confinement can be used as efficient adsorbents for environmental remediation processes. Nanomaterials span a variety of materials including inorganic, carbon-based, polymer, nanocomposites and biomaterials. In environmental remediation, inorganic nanomaterials have been widely applied [82]. Table 1 shows the characteristics and adsorption capacities of some inorganic nanoadsorbents for removal of Hg, Pb, Cd and As ions from solution.

## 2.1. Metal nanoparticles

Metal nanoparticles employed for the removal of heavy metals include Au, Ag and nano-zero-valent iron (n-ZVI) nanoparticles [7,104,105]. To improve the selectivity of some common adsorbents such as silica, alumina and activated carbon, Au and Ag metal nanoparticles have been used together with these materials to form composites or core–shell nanostructured materials [9,105,106]. Solis *et al.* [9] compared the adsorption capacities of a 3.19 nm size Au metal nanoparticle coated on silica and sand and found that the former showed a high affinity towards adsorption of Hg from water with $K_D$ (partition coefficient, which is a measure of sorbent affinity towards adsorbate) value of $9.96 \, l \, g^{-1}$ and adsorption efficiency of 96%. This confirmed the successful deposition of 6.9 mg of Au nanoparticles on a gram of silica compared with the sand which had only 1.5 mg deposition. The high selectivity was possible because Au forms a stable amalgam with Hg in the form of $Au_3Hg$. The $Au_3Hg$ alloy is thermodynamically favourable at room temperature, and thus, the mercury adsorbed was directly proportional to the Au content in the silica [104]. Other metals like Ag, Al and Cu also form amalgams with Hg [6,7,104]. The adsorption capacities of Au and Ag metal nanoparticles can be less than or equal to $800 \, mg \, g^{-1}$ which is comparable to metal oxides but may be relatively expensive [9,105].

## 2.2. Nano-zero-valent iron

Nano-zero-valent iron (n-ZVI) is a very promising metal nanoparticle that is gaining much interest as nanoadsorbents for a wide range of water pollutants including heavy metals [7]. The n-ZVI consists of an inner zero-valent iron $Fe^0$ core and an outer mixed iron oxides shell layer [7]. The core acts on the source of contaminants through electrostatic interaction and surface complexation [7]. The n-ZVI with a surface area of $26.3 \, m^2 \, g^{-1}$ and a wide range of particle sizes between 20 and 200 nm ensured the adsorption of $Cd^{2+}$ ions from an aqueous solution up to the tune of $769.2 \, mg \, g^{-1}$ [94]. Making a similar composite with activated carbon led to a low adsorption capacity of $142.8 \, mg \, g^{-1}$ for $Cd^{2+}$ [107]. Compositing 10 nm n-ZVI particle with graphene also ensured the removal of $Pb^{2+}$ from aqueous solution with an adsorption capacity of $585 \, mg \, g^{-1}$ [108]. In addition to the adsorption of $Pb^{2+}$, the n-ZVI reduces $Pb^{2+}$ to $Pb^0$, and this ensures the easy magnetic separation from solution [108]. However, there are possible handling challenges with n-ZVI, since it is highly reactive and may be toxic [109].

**Table 1.** Some nanoadsorbents were reported for the adsorption of Hg, Pb, Cd and As from an aqueous solution. LCMED: L-cysteine methyl ester dendrimer, HA: humic acid, Ga: glutaraldehyde, Cs: chitosan, CMC: carboxymethyl cellulose, L: SiO$_2$, lauric acid, oleic acid, multi-walled carbon nanotubes (MWCNTs), L-arginine, mesoporous silica, hydroxyapatite or ethylenediaminetetraacetic acid (EDTA), L-cyst.: L-cysteine, APTES: (3-aminopropyl)triethoxysilane, n-ZVI: nano-zero-valent iron, OPP: orange peel powder, Sd: sawdust, SdC: sawdust carbon, ZIF-8: zeolitic imidazolate framework-8, AA: ascorbic acid.

| adsorbent | surface area ($m^2\ g^{-1}$) | dose ($g\ l^{-1}$) | target heavy metal ion | metal ion conc. ($mg\ l^{-1}$) | pH | adsorption capacity ($mg\ g^{-1}$) | ref. |
|---|---|---|---|---|---|---|---|
| Al$_2$O$_3$-SiO$_2$-LCMED | 73.6 | 1 | Hg(II) | 5–4000 | 6 | 3232 | [71] |
| Fe$_3$O$_4$-HA | 64 | 0.1 | Hg(II) | 1 | 6 | 97.7 | [83] |
| Fe$_3$O$_4$-Ga-Cs | <3 | 1 | Hg(II) | 100 | 5–6 | 152 | [84] |
| CuO | 89.59 | 0.05 | Hg(II) | 10–100 | 9 | 825.21 | [85] |
| FeS-CMC | 36.9 | 0.0025 | Hg(II) | 4.8–40 | 6.5–10.5 | 3449 | [86] |
| γ-Fe$_2$O$_3$ | 79.35 | 10 | Pb(II) | 1–20 | 5 | 68.90 | [87] |
| γ-Fe$_2$O$_3$@L | 74–214 | 0.55 | Pb(II) | 40–50 | 7 | 49.30–88.2 | [88] |
| Fe$_3$O$_4$@L-cyst. | 58.49 | 2 | Pb(II) | 50 | 6 | 18.78 | [89] |
| SiO$_2$@APTES | 390 | 1 | Pb(II) | 100 | 3 | 40.40 | [90] |
| MnO$_2$-MWCNTs | 6.4 | 1 | Pb(II) | 10 | 7 | 20 | [91] |
| MgO | 72 | 0.67 | Pb(II) | 100 | 7 | 1980 | [92] |
| SnO$_2$ | 24.48 | 0.25 | Pb(II) | 100–400 | 4–7 | 1265.8 | [93] |
| n-ZVI | 26.3 | 0.5 | Cd(II) | 25–450 | — | 769.2 | [94] |
| Fe$_3$O$_4$@OPP | 65.19 | 0.2 | Cd(II) | 16 | 7 | 76.92 | [95] |
| Fe$_3$O$_4$-SdC@EDTA | 14 | 0.4 | Cd(II) | 30 | 6.5 | 63.30 | [96] |
| Fe$_3$O$_4$@Sd | 51.36 | 0.2 | Cd(II) | 50–250 | 7 | 1000 | [97] |
| SiO$_2$@APTES | 390 | 1 | Cd(II) | 100 | 6 | 49.46 | [90] |
| MgO | 72 | 0.67 | Cd(II) | 100 | 7 | 1500 | [92] |
| SnO$_2$ | 24.48 | 0.25 | Cd(II) | 100–400 | 4–7 | 1275.5 | [93] |
| ZnO | 8.25 | 0.4 | Cd(II) | 20–140 | 7 | 214.4 | [98] |
| ZrO$_2$ | 327 | ≤0.15 | As(III) | 0.3–100 | 7 | 83 | [99] |
| MnFe$_2$O$_4$ | 197.39 | 2 | As(V) | 10–400 | 2.1 | 68.25 | [100] |
| ZIF-8 | 1063.5 | 0.2 | As(V) | 20 | 7 | 60.03 | [101] |
| γ-Fe$_2$O$_3$ | 90.4 | 0.06 | As(V) | 1–11 | 3 | 50 | [102] |
| Fe$_3$O$_4$-AA | 179 | 0.06 | As(III) | 1 | 7 | 46.06 | [103] |

## 2.3. Magnetic metal oxide nanoparticles

The most widely used magnetic metal oxides are iron oxides. Their vast application as nanoadsorbents emanates from the possibility of rapidly separating them from solution using an external magnet [110]. Iron oxides including magnetite (Fe$_3$O$_4$), haematite (Fe$_2$O$_3$), maghaemite (γ-Fe$_2$O$_3$), goethite (α-FeO(OH)), lepidocrocite (γ-FeO(O)), manganese-doped iron oxide (MnFe$_2$O$_4$) and mixed iron oxides have been used for the adsorption of Pb$^{2+}$, Cd$^{2+}$ and Hg$^{2+}$ ions from aqueous solutions [88,109,111–113]. Bare iron oxides have shown adsorption capacities ranging 1.69–101.1 and 0.1105–820.16 mg g$^{-1}$ for Cd$^{2+}$ and Pb$^{2+}$, respectively [109,112,114–116]. Goethite and lepidocrocite as two different phases of iron oxide/hydroxide have shown high adsorption capacities of 820.16 and 527.94 mg g$^{-1}$, respectively, for Pb$^{2+}$ due to the OH surface groups on their surfaces which bind strongly with Pb$^{2+}$ [112]. Manganese-doped iron oxide (MnFe$_2$O$_4$) nanoparticles also have been shown to possess high adsorption

capacities of 488, 97 and 136 mg g$^{-1}$ for Pb$^{2+}$, As$^{3+}$ and As$^{5+}$, respectively [117]. The use of Fe$_3$O$_4$@C@MnO$_2$ for removal of Eu(III)/U(VI), X-ray photoelectron spectroscopy and zeta potential analyses showed that the adsorption process was governed by surface complexation and electrostatic attraction [118].

The superparamagnetic nature of iron oxides coupled with the nanoscale capabilities makes them excellent adsorbent materials. In addition, iron oxide surfaces in many other studies were functionalized with suitable functional groups aimed at improving their selectivity towards targeted pollutants. Lauric acid, oleic acid, L-arginine, ethylenediaminetetraacetic acid, sulfo (SO$_3$H), maleate, humic acid (HA), polyrhodamine and 2-mercapto benzothiazole-capped iron oxide nanoparticles have been used for the removal of Cd$^{2+}$, Pb$^{2+}$ and Hg$^{2+}$ from aqueous solution [78,88,110,119,120]. These functionalized iron oxides afforded an adsorption capacity range of 0.59–108.93 mg g$^{-1}$. Compositing iron oxide with silica or chitosan with further functionalization with thiol and glutaraldehyde resulted in Hg$^{2+}$ adsorption capacities of 148.8 and 152 mg g$^{-1}$, respectively [84,121]. Interestingly, Fe$_3$O$_4$ nanoparticles coated with natural biodegradable materials such as sawdust, orange peel powder, cashew nutshell resin and shellac (natural resin with abundant hydroxyl and carboxylic groups) have been used for the adsorption of Cd$^{2+}$ [95–97,122,123]. Sawdust-coated Fe$_3$O$_4$ nanoparticle with BET active surface area of 51.36 m$^2$ g$^{-1}$ incredibly removed Cd$^{2+}$ from 50 to 250 mg l$^{-1}$ Cd$^{2+}$ aqueous solution, and the adsorption capacity was reported to be 1 g g$^{-1}$ [97]. However, Fe$_3$O$_4$ coated with shellac, orange peel powder, and sawdust carbon showed relatively lower Cd$^{2+}$ adsorption capacities of 18.8, 76.92 and 63.3 mg g$^{-1}$, respectively [95,96,122]. Irrespective of the various functionalization treatments, iron oxides maintained their magnetic properties which enable them to be separated from the solution.

## 2.4. Other metal oxide nanoparticles

Apart from iron oxide, nanoparticles of metal oxides such as silica (SiO$_2$), alumina (Al$_2$O$_3$), magnesium oxide (MgO), caesium oxide (CeO$_2$), titanium dioxide (TiO$_2$), tin oxide (SnO$_2$), zinc oxide (ZnO), manganese dioxide (MnO$_2$), copper oxide (CuO), nickel oxide (NiO$_2$) and zirconium oxide (ZrO$_2$) have been used as nanoadsorbents for sequestration of heavy metals from water. Table 1 shows adsorption conditions and capacities of different metal oxide nanoadsorbent for Hg$^{2+}$, Cd$^{2+}$, Pb$^{2+}$ and As$^{3+}$.

Amorphous silica is non-toxic, has a high specific surface area and regular pore structures, making it a very promising adsorbent material [73]. Aminopropyl-functionalized silica nanoparticles with an average size of 7 nm possessed a large BET surface area of 390 ± 40 m$^2$ g$^{-1}$ and could adsorb 0.44 (49.46) and 0.195 (40.40) mmol g$^{-1}$ (mg g$^{-1}$) of Cd$^{2+}$ and Pb$^{2+}$, respectively [90]. Arce *et al.* [90] confirmed using DFT quantum computational calculations the formation of Pb-N and Pb-C bond distances which strongly agree with the extended X-ray absorption fine structure data. Alumina with a particle size of approximately, 6–13 nm adsorbed Pb$^{2+}$ at 47.08 mg g$^{-1}$ capacity [124]. Mixed oxides of silica and alumina nanoparticles functionalized with L-cysteine methyl ester dendrimer demonstrated an incredible adsorption capacity of 3232 mg g$^{-1}$ for Hg$^{2+}$ [71]. Figure 1 shows the morphology and performance of the L-cysteine methyl ester dendrimer-coated mixed oxides of silica and alumina nanoparticles. Alumina-silicate-mixed oxides are resistant to oxidation, chemically and thermally stable and have high conductivity, low dielectric constant, creep resistant and low thermal expansion [71].

Nanoparticles of CuO and ZnO also have high adsorption capacities for Hg$^{2+}$. Fakhri [85] and Sheela *et al.* [126] reported CuO and ZnO adsorption capacities of approximately 825.21 and 714 mg g$^{-1}$, respectively, for Hg$^{2+}$. The high adsorption of CuO may be that Cu forms a stable amalgam with Hg$^{2+}$ [104]. SnO nanoparticles are conducting, transparent and gas-sensitive, and thus, are gaining much interest as nanoadsorbents and as catalysts. SnO with a surface area of 24.48 m$^2$ g$^{-1}$ as measured by BET isotherms demonstrated adsorption of Pb$^{2+}$ and Cd$^{2+}$ at 1265.8 and 1275.5 mg g$^{-1}$, respectively [93]. Flowerlike MgO nanostructures also displayed excellent adsorption of Pb$^{2+}$ and Cd$^{2+}$. Figure 2 shows the TEM images and adsorption isotherms of the flowerlike MgO nanostructures. With a BET surface area of 72 m$^2$ g$^{-1}$, MgO was able to remove Pb$^{2+}$ and Cd$^{2+}$ ions from 100 mg l$^{-1}$ metal ions solution at adsorption capacities of 1980 and 1500 mg g$^{-1}$, respectively [92]. The adsorption mechanism exhibited by MgO is ion exchange, leaving some level of Mg$^{2+}$ in solution. However, Mg$^{2+}$ is not toxic and the recommended level in drinking water is 450 mg l$^{-1}$ according to WHO [92]. We note that there appear to be very few reports on the use of CuO, ZnO, SnO and MgO nanoparticles for heavy metal sequestration. Though they possess impressive heavy metals adsorption capacities, their removal from the solution will still be a challenge. To impart magnetization to these materials, in order to ensure simplicity of removal from solution using an

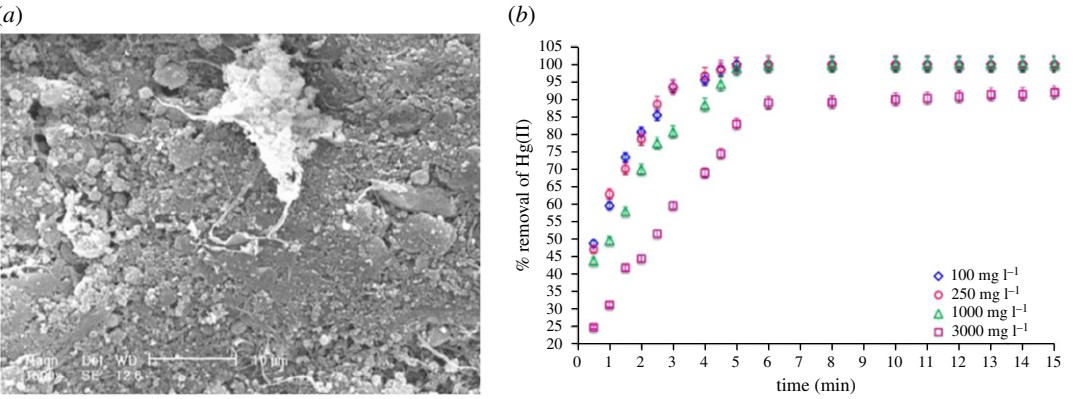

**Figure 1.** (*a*) SEM images of mixed oxides of silica and alumina nanoparticle functionalized with L-cysteine methyl ester dendrimer and (*b*) its adsorption kinetics for the adsorption of Hg(II) ions at different concentrations. Reprinted with permission from Arshadi *et al.* [71]. Copyright (2021) Elsevier.

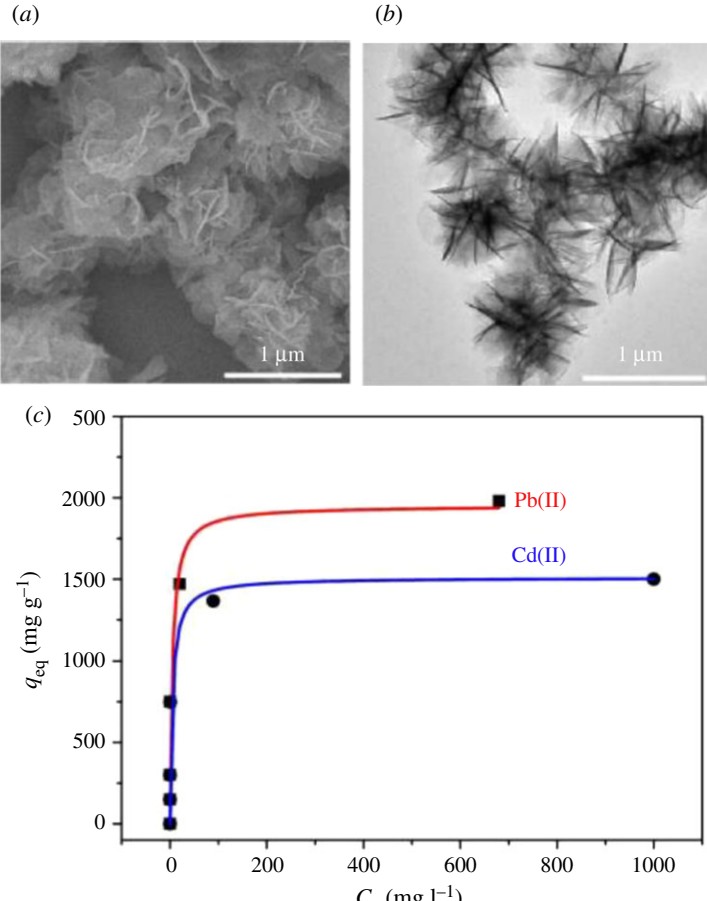

**Figure 2.** TEM image (*a*) low-magnification and (*b*) high-magnification of flowerlike MgO nanostructures, and (*c*) Pb(II) and Cd(II) adsorption isotherms obtained using flowerlike MgO nanostructures. Reprinted with permission from Cao *et al.* [92]. Copyright (2021) American Chemical Society.

external magnet, doping with Fe and Mn could potentially be a strategy to impart these magnetic properties to the host material.

## 2.5. Metal sulfide nanoparticles

Compared with metal oxides, metal sulfides often possess a superior affinity for heavy metal ions; however, they have received less attention as nanoadsorbents for heavy metal removal from water

[128]. Metal sulfides remove divalent metal ions through (i) chemical precipitation (equation (2.1)), (ii) ion exchange (equation (2.2)), and (iii) complexation (equation (2.3)) [86]. The most studied metal sulfide as adsorbent for heavy metals removal is iron sulfide (FeS) or mackinawite.

$$MS(s) + A^{2+}(aq) \rightarrow AS(s) + M^{2+}(aq), \tag{2.1}$$

$$MS(s) + xA^{2+}(aq) \rightarrow [M_{(1-x)}, A_x] S(s) + xM^{2+}(aq), \quad (x < 1) \tag{2.2}$$

and

$$MS(s) + A^{2+}(aq) \rightarrow MS - - - - - A^{2+}. \tag{2.3}$$

FeS naturally scavenges for $Hg^{2+}$ and forms stable $\beta$-HgS (metacinnabar, with solubility product, $K_{sp}$, $4 \times 10^{-54}$) even in the presence of organic matter [86]. FeS also removes divalent metal ions such as $Mn^{2+}$, $Ca^{2+}$, $Mg^{2+}$, $Ni^{2+}$ and $Cd^{2+}$. Additionally, FeS is a reductant and acts as an electron donor, making it possible to reduce and detoxify chlorinated organic compounds and inorganic oxyanions of Cr(VI), Se(VI) and As(VI) [129]. Remarkably, carboxymethyl cellulose (CMC)-stabilized FeS offered an $Hg^{2+}$ sorption capacity of 3449 mg g$^{-1}$, and this was attributed to the dual-mode sorption mechanism where CMC-FeS adsorbs $Hg^{2+}$ through concurrent precipitation [86]. Similarly, a large $Pb^{2+}$ remediation capacity of 2950 mg g$^{-1}$ has been demonstrated by Pala & Brock [128] with zinc sulfide (ZnS) gel, where the mechanism is mainly more of ion exchange than adsorption. ZnS has been successfully used to sequentially remove $Hg^{2+}$, $Cu^{2+}$, $Pb^{2+}$ and $Cd^{2+}$ from simulated contaminated water containing 5–678 mg l$^{-1}$ of heavy metal ions at removal efficiencies of 99.9%, 99.9%, 90.8% and 66.3%, respectively [77]. The ZnS showed higher selectivity towards $Hg^{2+}$ and $Cu^{2+}$ than $Pb^{2+}$ and $Cd^{2+}$, and this Fang *et al.* [77] attributed to the differences in the solubility product ($K_{sp}$) of the metal sulfides. The lower the $K_{sp}$, the higher the stability of the sulfide, and hence, the higher the precipitation and adsorption. The selectivity of the sulfides may also be attributed to the hard–soft acids and bases (HSAB) theory pioneered by Pearson (vide infra), where the sulfur which is a soft base will have a higher affinity for $Hg^{2+}$ which is a soft acid. However, adequate research will be required to clearly establish the mechanism of metal sulfides selective adsorption.

Core–shell nanostructured materials, such as $Fe_3O_4$-ZnS combines the magnetic properties of the iron oxide and the better affinity of ZnS, making sure that adsorbent material is easily separable and also selective. The magnetic $Fe_3O_4$-ZnS nanoparticles gave an adsorption capacity of 129.9 mg g$^{-1}$ for removal of $Hg^{2+}$ from water which is better compared with non-magnetic adsorbents [130].

The different nanomaterials discussed above for heavy metals sequestration have unique advantages and disadvantages (table 2). Thus, a search for new inorganic nanoadsorbents with efficient and robust properties or qualities is desired.

# 3. Qualities of nanoadsorbents

The qualities of nanoparticles that make them suitable as nanoadsorbents for sequestration of heavy metals include (i) simple and cheap synthesis pathway, (ii) selectivity and affinity towards the target pollutant, (iii) thermal and chemical stability in solution, (iv) ease of removal from solution after adsorption, (v) it should be easily regenerated and re-used severally, and (vi) should have the ability to perform other functions apart from adsorption.

## 3.1. Synthesis pathway

The synthetic pathways generally reported for nanoadsorbent synthesis usually include (i) reduction, (ii) co-precipitation, (iii) sol–gel, and (iv) hot-injection methods. Figure 3 shows the different synthesis pathways for nanoadsorbent synthesis.

### 3.1.1. Reduction

The reduction route is often employed for the synthesis of metal nanoparticles such as Au, Ag and n-ZVI. Ojea-Jiménez *et al.* [104] synthesized Au nanoparticles (with size $8.9 \pm 1.6$ nm) by reducing $Au^{3+}$ with sodium citrate ($Na_3C_6H_5O_7$) and was able to achieve 100% of Hg removal from water; however, the initial concentration of Hg was only 0.16 ppm. Similarly, Lisha and Pradeep [132] synthesized Au nanoparticles by using $Na_3C_6H_5O_7$ to reduce $Au^{3+}$ to $Au^0$ nanoparticles and further supported it with alumina, resulting in Hg adsorption capacity of 4.065 g g$^{-1}$. Sodium borohydride ($NaBH_4$) has also been used to reduce $Ag^+$ to $Ag^0$ nanoparticles for adsorption of Hg [105]. Using $NaBH_4$, Yan *et al.* [7]

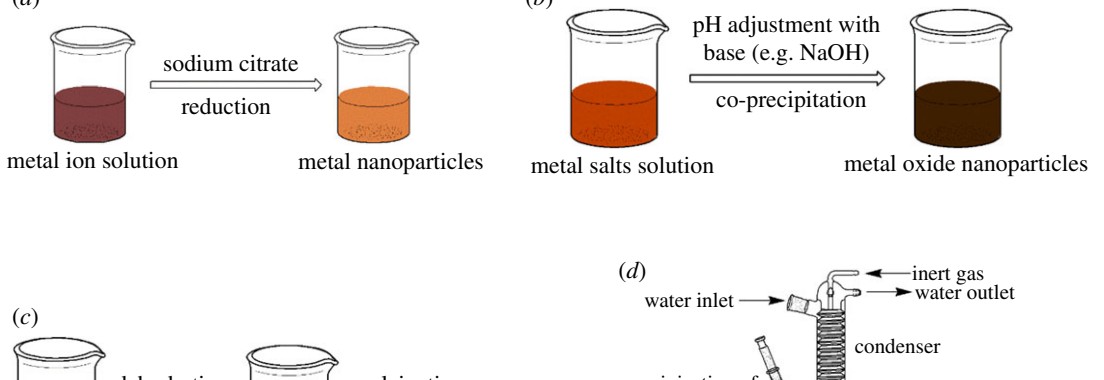

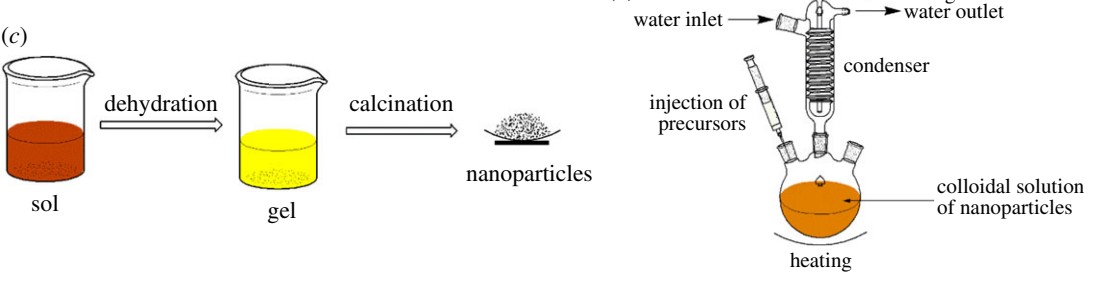

**Figure 3.** Synthesis pathways for nanoadsorbents: (*a*) reduction, (*b*) co-precipitation, (*c*) sol–gel and (*d*) hot-injection methods.

**Table 2.** Advantages and disadvantages of nanoadsorbent materials.

| nanoparticle | advantages | disadvantages | ref. |
|---|---|---|---|
| metal | – forms stable amalgam with heavy metal pollutant (e.g. Hg) | – relatively expensive | [9,105] |
| n-ZVI | – adsorbs wide range of water pollutants | – has handling challenges | [109] |
| | – possesses reductive potential | – may be toxic | |
| | – highly reactive | | |
| magnetic metal oxide | – rapid separation from aqueous solution using an external magnet | – has relatively low adsorption capacity for heavy metals | [87,97] |
| | | – requires functionalization to improve adsorption capacity | |
| non-magnetic metal oxide | – has high adsorption capacity for heavy metals (e.g. CuO, ZnO, SnO, MgO) | – possible challenges with separation from aqueous solutions | [71,92] |
| metal sulfide | – possesses specific affinity for heavy metals adsorption | – prone to oxidation | [131] |

synthesized n-ZVI nanoparticles, which were able to adsorb 98% of Hg from 40 mg l$^{-1}$ Hg solution within 2 min. Sodium citrate and NaBH$_4$ are efficient reducing agents commonly used in nanoparticles synthesis. While sodium citrate is food grade and biologically compatible, NaBH$_4$ introduces boron into the water which could raise public health concerns.

### 3.1.2. Co-precipitation

In co-precipitation synthesis, the solutes are usually soluble at a particular condition but precipitate out of solution when supersaturation is reached. Co-precipitation is simple and rapid, provides easy control of the particle size and composition, provides the possibility of nanoparticle surface modification, is energy-efficient, and does not require organic solvents. However, co-precipitation could yield impure nanoparticles with irregular and less homogenized nanoparticle sizes; it is time-consuming and has poor reproducibility [133]. Fe$_3$O$_4$ has been synthesized widely using the co-precipitation technique, largely due to its simplicity and easiness of modification of the nanoparticle surface. Mehdinia *et al.* [2] synthesized Fe$_3$O$_4$ nanoparticles using co-precipitation, supported it with silica and functionalized the surface with dithiocarbamate, which resulted in approximately 99% adsorption of Hg from water.

Similarly, $Fe_3O_4$ nanoparticles have been synthesized via co-precipitation and functionalized with glutathione, glutaraldehyde, thiol, polyrhodanine and HA and applied for Hg adsorption from water, affording adsorption capacities between 34.48 and 152 mg g$^{-1}$ [79,83,84,121,134].

### 3.1.3. Sol–gel synthesis

The sol–gel method employs solvents and chemical reagents that after hydrolysis and condensation, the mixture of solvents turns into a gel. Arshadi *et al.* [71] synthesized $SiO_2$–$Al_2O_3$-mixed oxide nanoparticles using the sol–gel method. Typically, the chemical reagents: aluminium tri-sec-butylate and tetraethyl orthosilicate were dissolved in n-butanol and the resultant solution warmed at 60°C. Acetylacetone (H-acac) was added slowly to the solvent mixture after cooling to room temperature to obtain a clear solution. This clear solution was hydrolysed by adding an alkoxide solution and left overnight, resulting in the formation of a transparent gel. The gel was dried to remove the solvents and finally calcined at 500°C to remove the other organic components to obtain the nanoparticles. This nanomaterial was reported to have removed approximately 99.9% of Hg from water with a very high initial Hg concentration of 500 mg l$^{-1}$. Fakhri [85] also using the sol–gel technique, synthesized CuO nanoparticles which had an Hg adsorption capacity of approximately 825.21 mg g$^{-1}$. The nanomaterials obtained from the sol–gel method are usually of high purity and porous. Though the sol–gel method gives control over porosity, particle size, chemical composition and dopant incorporation, compared with the other methods, it usually requires longer reaction time and involves organic solvents and higher temperatures to remove the organic components from the gel [133].

### 3.1.4. Hot-Injection

The hot-injection method involves the injection of a solution of reagents into hot solvents, which gives control over the nucleation and growth of the nanoparticles. It is usually employed for the synthesis of metal chalcogenide nanocrystals. ZnS supported on $Al_2O_3$ was synthesized using the hot-injection method by Fang *et al.* [77] for the sequential removal of $Hg^{2+}$, $Cu^{2+}$, $Pb^{2+}$, $Cd^{2+}$ and $Zn^{2+}$ in water. The ZnS-$Al_2O_3$ was prepared by mixing α-$Al_2O_3$ and $ZnCl_2$ in ethylene glycol in a three-necked flask fitted with a condenser, and maintaining the temperature at 180°C, 1-butylamine solution of thiourea was injected into it continuously, and the resulting mixture aged for 3 h. Although the hot-injection method is efficient and ensures that good quality monodispersed nanoparticles are obtained, it requires a complex set-up and organic solvents, and the resulting nanoparticles' surfaces are already occupied by capping groups or solvents, which makes it unattractive as nanoadsorbents.

Table 3 shows the advantages and disadvantages of the different synthetic methods that have been used for nanoadsorbents syntheses. It must be noted that very few synthetic methods have been explored for the synthesis of nanoadsorbents, and admittedly, this research gap makes it challenging in deciding on suitable synthetic approaches for nanoadsorbents for heavy metals remediation.

## 3.2. Selectivity and affinity

Activated charcoal is a widely known efficient adsorbent. This is because it is cheap, available and has a high capacity for the removal of a wide range of pollutants. However, activated charcoal is a non-selective adsorbent and has no special affinity towards any specific pollutant. Affinity facilitates the rate of adsorption [9]. This makes nanoadsorbents superior to activated charcoal and other non-specific adsorbents because they could be tailored to possess special affinity and selectivity towards a particular pollutant. Figure 4 shows the chemical structures of compounds used for the functionalization of the nanoparticle surface for improving selectivity towards heavy metals in solution. To improve the preferential adsorption of $Hg^{2+}$ onto $SiO_2$–$Al_2O_3$-mixed oxide nanoparticles, Arshadi *et al.* [71] covalently immobilized L-cysteine methyl ester dendrimer onto the surface of the nanoparticle and the selectivity was evaluated in the presence of $Pd^{2+}$, $Pb^{2+}$, $Cd^{2+}$, $Zn^{2+}$, $Cu^{2+}$, $Co^{2+}$ and $Mn^{2+}$. At an initial concentration of 250 mg l$^{-1}$ of metal ions, the functionalized nanoparticles exhibited high selectivity for adsorption of $Hg^{2+}$ of about 96.2%, and low adsorption for the interfering cations at adsorption capacities of 10, 3.7, 1.13, 0.76, 5.81, 1.84 and 3.02%, respectively. However, when the metal cations concentration was increased to 1000 mg l$^{-1}$, the adsorption of $Hg^{2+}$ decreased to 90%. The selective adsorption was attributed to the –SH and –NH functional groups of the L-cysteine methyl ester which preferentially donated electrons to the $Hg^{2+}$ resulting in complexation [71].

**Table 3.** Advantages and disadvantages of synthesis methods for nanoadsorbents.

| synthesis method | advantages | disadvantages | ref. |
|---|---|---|---|
| reduction | – simple | – some reducing agents may be toxic | [105] |
| | – suitable for the synthesis of metal nanoparticles | | |
| co-precipitation | – simple and rapid | – yields impure nanoparticles with irregular and less homogenized particle sizes | [133] |
| | – provides control of particle size and composition | – time-consuming | |
| | – provides a possibility for nanoparticle surface functionalization | – poor reproducibility | |
| | – energy efficient | | |
| | – requires no organic solvents | | |
| sol–gel | – yields high purity and porous nanoparticles | – time-consuming | [71,133] |
| | – provides control over porosity, particle size, composition and dopant incorporation | – involves the use of organic solvent | |
| | | – requires high temperature to remove the organic components from the gel | |
| hot-injection | – provides control over nucleation and growth of particles | – requires complex set-up | [77] |
| | – efficient | – requires organic solvents and capping agents | |
| | – yields monodispersed nanoparticles | – particle surface always covered with capping groups | |

Zhang *et al.* [121] demonstrated that thiol-functionalized $Fe_3O_4@SiO_2$ could selectively remove $Hg^{2+}$ at an adsorption capacity of 110 mg g$^{-1}$ even in the presence of $K^+$, $Na^+$ and $Ca^{2+}$ cations which naturally abound in water at high concentrations. While the hydroxyl groups on the surface of the $Fe_3O_4@SiO_2$ nanoparticle are hard Lewis bases and, therefore, preferred bonding with hard Lewis acids such as alkaline metals and alkaline earth metals, the thiol group being soft Lewis base exhibited a strong preference for bonding with $Hg^{2+}$ which is a soft Lewis acid. Mehdinia *et al.* [2] compared thiol and amine-functionalized magnetic mesoporous silica nanoparticles and found that the thiol functional group was more selective towards $Hg^{2+}$ removal than the amine. Mercury adsorption capacity of 538.9 mg g$^{-1}$ was achieved with the thiol-functionalized nanoparticles, and this was attributed to the thiol functional group. In the presence of competitive cations, the thiol-functionalized nanoparticles were selective to the cations in the order $Hg^{2+} > Pb^{2+} > Cd^{2+} > Zn^{2+}$. This trend was also confirmed by Košak *et al.* [73] when they found mercaptopropyl-functionalized $SiO_2$ nanoparticles to adsorb 99.9% $Hg^{2+}$, 55.9% $Pb^{2+}$, 50.2% $Cd^{2+}$ and 4% $Zn^{2+}$. Unfortunately, the selectivity of the amine-functionalized nanoparticles was not tested in the presence of the competing cations.

The binding interactions between the carboxylic acid functional groups in free molecules of 2-mercato-4-methyl-5-thiazoleacetic acid (MCT), monomercaptosuccinic acid (MMSA), o-thiosalicylic acid (o-TSA) and p-thiosalicylic acid (p-TSA), and $Hg^{2+}$, $Pb^{2+}$ and $Cd^{2+}$ cations were explored by Hamid *et al.* [74]. These molecules also contain thiol functional group (figure 4). They realized that the COOH in MCT binds completely (100%) with the $Pb^{2+}$ and $Cd^{2+}$, and only 20% with $Hg^{2+}$; the COOH in MMSA binds 97% with $Pb^{2+}$, 91% with $Cd^{2+}$ and had no interaction with $Hg^{2+}$; the COOH

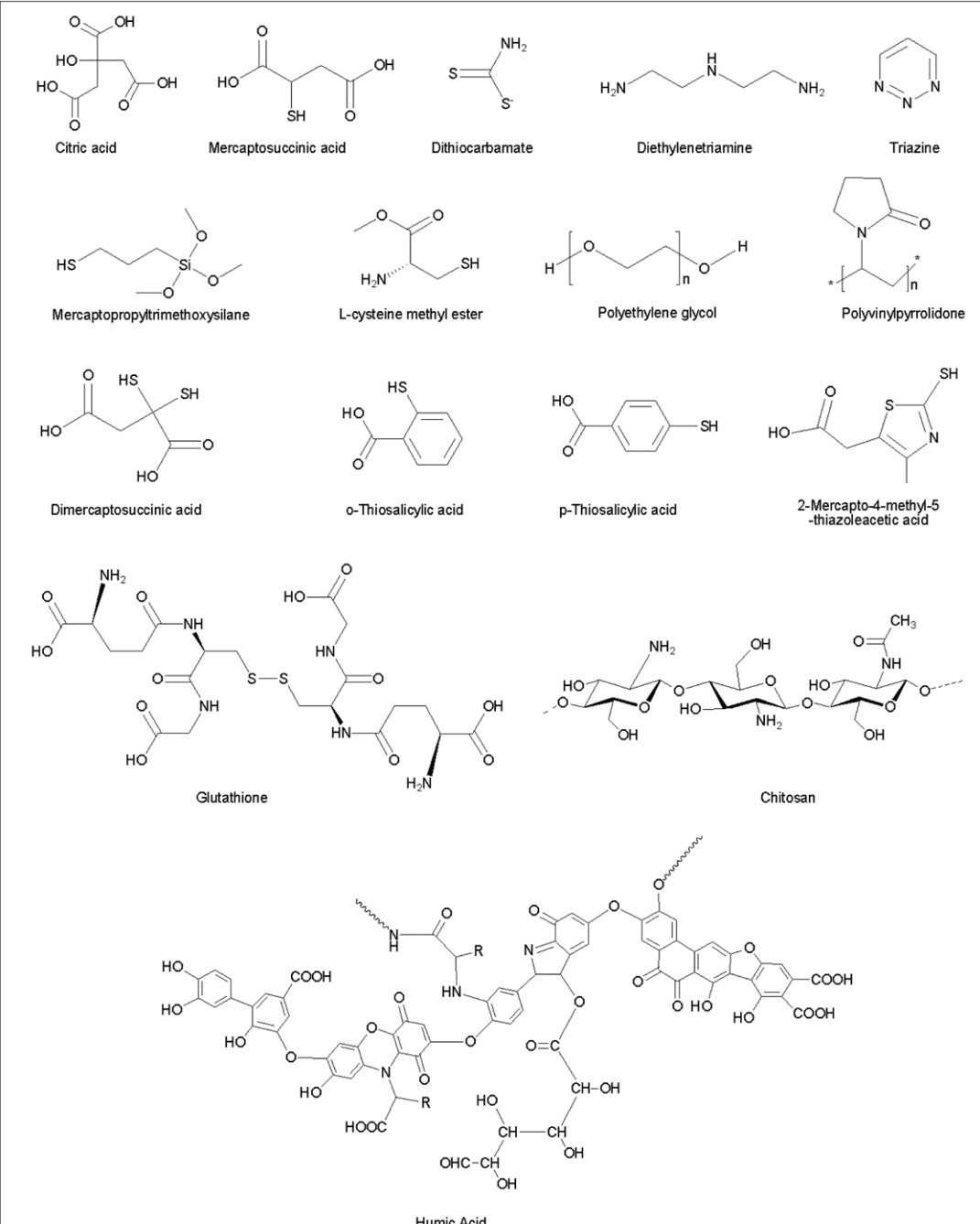

**Figure 4.** Molecules reported as capping agents for coating nanoparticle surfaces for improving their affinity and selectivity towards mercury adsorption in water.

in o-TSA binds 96% with $Pb^{2+}$, 90% with $Cd^{2+}$ and 3% with $Hg^{2+}$; the COOH in p-TSA binds 69% with $Pb^{2+}$, 65% with $Cd^{2+}$ and 7% with $Hg^{2+}$. It appeared that the strength of the interaction of COOH in any given molecule follows the order Pb > Cd > Hg. However, when silica nanoparticles were functionalized with these molecules through amide linkage using the COOH groups, the selectivity for Hg adsorption was improved. This was because the COOH group which is selective towards $Pb^{2+}$ and $Cd^{2+}$ was blocked or used in the amidation reaction. The projected SH groups were said to be responsible for $Hg^{2+}$ selective adsorption. Unfortunately, the adsorption of the $Hg^{2+}$ was found to be affected by the steric hindrance of the capping molecules attached to the nanoparticle surface [74].

The fundamental principle that may explain the affinity and selectivity of functionalized nanoadsorbents is Pearson's theory of hard and soft acids and bases—HSAB [73,121]. HSAB theory states that a soft acid would prefer to coordinate and form stronger bonds and more stable complexes

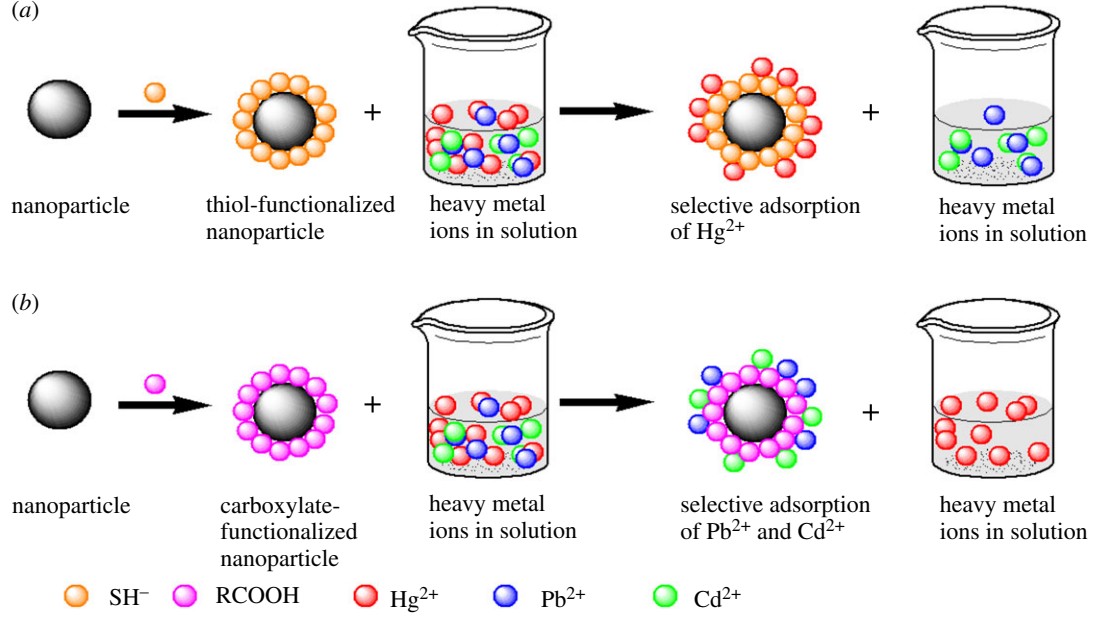

**Figure 5.** Selective adsorption of heavy metal ions in a solution based on the Pearson's theory of hard and soft acids and bases using inorganic nanoparticle (*a*) thiol functionalized nanoparticle selectively adsorbs $Hg^{2+}$ and (*b*) carboxylate functionalized nanoparticle selectively adsorbs $Pb^{2+}$ and $Cd^{2+}$.

with soft bases, whereas a hard acid would prefer to coordinate and form stronger bonds and more stable complexes with hard bases [135]. Hard acids are relatively small metal ions that are highly electronegative and have lower polarizabilities, whereas soft acids are relatively large metal ions that are less electronegative and have high polarizabilities [135]. $Hg^{2+}$ and $Cd^{2+}$ are classified as soft acids and usually form stable complexes with soft bases such as SH [135]. COOH and $RNH_2$ are hard bases and strongly bind with hard acids such as $Fe^{3+}$, $Sn^{2+}$, $Cr^{3+}$ and $As^{3+}$ [135]. $Pb^{2+}$, $Ni^{2+}$, $Fe^{2+}$, $Cu^{2+}$ and $Zn^{2+}$ are classified as borderline acids and may prefer to bind with hard or soft bases [135]. Figure 5 shows the binding preferences of COOH and SH groups towards metal ions based on the HSAB theory. The selective adsorptivity of nanoadsorbents towards $Hg^{2+}$ can be enhanced by functionalizing the surface of the nanoparticle with thiol-rich molecules, whereas attaching carboxylic containing molecules will improve adsorption of $Pb^{2+}$. However, bulky molecules may pose steric restrictions to the approaching cation. The process of nanoparticle surface functionalization can be complicated. Surface functionalization may lead to the reduction of the active surface area of the nanoadsorbents (electronic supplementary material, table S1).

Mehdinia *et al.* [2] observed the decrease of silica nanoparticle surface area by approximately 31% when it was functionalized with thiol groups. However, HA did not reduce the surface area of $Fe_3O_4$ [83]. It must be stated that the HSAB classification was made based on the equilibrium constants of an acid–base adduct or complex molecule in solution [135]. Fang *et al.* [77] confirms HSAB by attributing the removal efficiency of heavy metals from solution to the solubility products ($K_{sp}$) of their sulfides, that is, the lower the $K_{sp}$ of the sulfide of the heavy metal, the higher its removal efficiency. However, the effect of $K_{sp}$ on selective adsorption has not been widely studied.

## 3.3. Regeneration, reusability and stability

The pollutants chemisorb on the surface of the nanoadsorbent, which implies that chemical bonds are formed between the nanoadsorbent and the pollutant. A suitable nanoadsorbent should not have a complicated desorption process. The adsorbed pollutants should be easily desorbed into the solution and the adsorbent re-used. Adsorbents with high recycling or regeneration potential are of greatest interest. Acids, bases, thermal treatment and amalgamation have been used in heavy metals desorption processes. $Hg^{2+}$ was desorbed from a dithio-functionalized magnetic silica nanoparticle by treating with a mixture of 1 M $HNO_3$ and 2% thiourea, and the recovery efficiency of the $Hg^{2+}$ was greater than 98% after three concurrent uses. The thiourea acted as a chelating agent which complexed with the $Hg^{2+}$ to detach it from the silica nanoparticle surface [2]. Iodine ions also possess

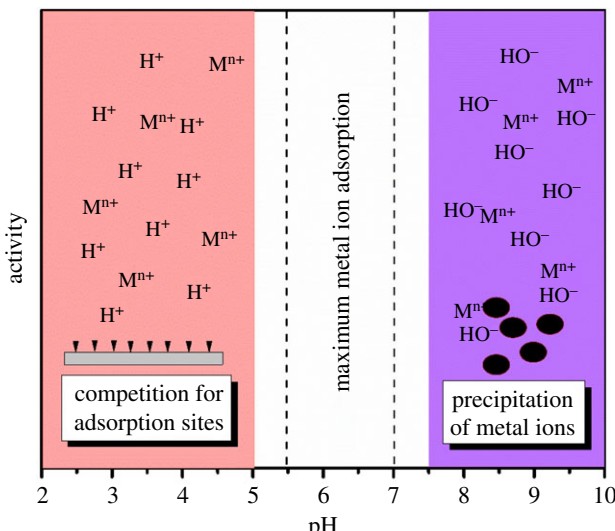

**Figure 6.** Sketch of a graph showing adsorption characteristics of heavy metal ions on inorganic nanoadsorbents at different pH ranges.

a strong affinity for $Hg^{2+}$ ions, making potassium iodide a suitable recovery agent for detaching $Hg^{2+}$ from adsorbents [134].

The ability of a nanoadsorbent to withstand the desorption processes during regeneration or its resilience in different solvent media determines its stability. $Hg^{2+}$ adsorbed L-cysteine methyl ester dendrimer-capped mixed oxide nanoadsorbent was regenerated using sequential solutions of EDTA, 0.15 M HCl and water, dried at 80°C and re-used more than eight times to achieve adsorption capacity of greater than 97.5% with only a loss of 7% of the dendrimer [71]. $Fe_3O_4$ nanoparticles are susceptible to air oxidation and tend to aggregate in aqueous solutions [83,111,121]. Coating bare $Fe_3O_4$ nanoparticles with HAs improves their stability as nanoadsorbent. Liu *et al.* [83] soaked bare $Fe_3O_4$ and HA-coated $Fe_3O_4$ nanoparticles in water for 30 days and observed that the bare $Fe_3O_4$ easily oxidized into brown suspension and lost its magnetization, leaching $0.24$ mg l$^{-1}$ of free iron ions in solution, whereas the HA-$Fe_3O_4$ maintained its magnetization, and only $0.025$ mg l$^{-1}$ of free iron ions and $0.16$ mg l$^{-1}$ organic carbon were leached into solution. Zhang *et al.* [121] coated $Fe_3O_4$ with silica which improved its chemical stability, biocompatibility and versatility in surface modification. Similarly, Nasirimoghaddam *et al.* [72] coated $Fe_3O_4$ with chitosan, a natural polyaminosaccharide, and were able to obtain $Hg^{2+}$ adsorption capacity of 90.58%, whereas the uncoated $Fe_3O_4$ only removed 41%.

The rate of adsorption of heavy metal ions is dependent on the pH of the solution. At very low pH values, the high concentration of $H^+$ ions compete with the metal ions for available active sites of the adsorbent material, whereas at high pH values, the $OH^-$ ions tend to precipitate the metal ions, preventing their adsorption. The maximum adsorption of $Hg^{2+}$ has been achieved within pH values of 5–7.5 (figure 6) [8,75,105,134]. Chitosan-bound $Fe_3O_4$ nanoparticles were able to remove 92.4% of $Hg^{2+}$ ions even at a low pH of 3 [72]. Similarly, 2-mercaptobenzothiazole-coated $Fe_3O_4$ nanoparticles $Hg^{2+}$ removal efficiency was not significantly affected within a wide pH range of 2.5–11 [78]. The chitosan and 2-mercaptobenzothiazole coatings provided adequate stability to the $Fe_3O_4$ nanoparticles, thus, making them robust nanoadsorbents. Coating of nanoadsorbents seems a better option for improving the stability in solution and during desorption processes; however, the synthetic process of functionalization of nanoparticle surface remains a major challenge.

In addition, the information on the point of zero charge ($pH_{pzc}$) of nanoadsorbents is relevant in determining the pH at which the net surface charge is zero, which also ensures the robustness of the adsorbents in different media. The $pH_{pzc}$ indicates the mechanism of adsorption (whether cationic or anionic) and helps in selecting the appropriate pH at which the adsorption process should be carried out in order to minimize precipitation. However, there is little information on $pH_{pzc}$ of various nanoadsorbents in the literature. Electronic supplementary material, table S2 shows the $pH_{pzc}$ of some nanoadsorbents. The surface capping characteristics have a significant effect on the $pH_{pzc}$ of the adsorbent. The $pH_{pzc}$ of bare $Fe_3O_4$ in 0.01 M NaCl solution was 6.78 and when coated with orange peel powder, reduced to 5.21 [95]. Similarly, $Fe_3O_4$ nanoparticles capped with sawdust carbon and

L-cysteine had $pH_{pzc}$ values of 4.3–6.1 and 5.7, respectively [89,97]. Capping $SiO_2$ nanoparticles with APTES ((3-aminopropyl) triethoxysilane) significantly increased the $pH_{pzc}$ from 2.5 to 9.15, indicating improved robustness of the nanoadsorbent in a highly basic environment [88]. However, some material such as ZnO is preferred as nanoadsorbent because it possesses a wide $pH_{pzc}$ value of 8–9 [126].

## 3.4. Removal of adsorbent from solution

The current trend of nanoadsorbents possesses some level of magnetization which allows them to be easily removed from the solution by applying an external magnet [72]. This makes magnetic nanoadsorbents superior to other non-magnetic adsorbents such as activated charcoal. $Fe_3O_4$ nanoparticles maintained magnetization at saturation of 79.6 emu $g^{-1}$ even after being coated with HA and only require a low magnetic field gradient for separation from the solution [83]. Non-magnetic inorganic nanoadsorbents can be doped with Fe, Mn and Co ions to impact some level of magnetization; however, very few works have been done using doped nanoparticles as adsorbents.

## 3.5. The ability of adsorbent to perform other functions

Apart from the adsorption process, some nanoadsorbents have the potential to perform other functions such as amalgamation. Gold forms amalgam with mercury; however, it requires that mercury is in the reduced form $Hg^0$ before the amalgamation could occur. Not all nanoadsorbents form stable amalgams with the target pollutant. In most instances, $Hg^{2+}$ ions are the predominant form of mercury pollution in water. Ojea-Jiménez et al. [104] developed an Au nanoparticle coated with citrate ions which together with its adsorption capabilities was able to reduce $Hg^{2+}$ to $Hg^0$ which resulted in the formation of amalgam with the Au nanoparticle ($Au_3Hg$). Thus, the use of very strong and toxic-reducing agents such as $NaBH_4$ was avoided. This observation makes nanoparticles of metals such as silver, copper, tin and aluminium, which form a stable amalgam with Hg attractive and promising nanoadsorbents for sequestration of mercury pollution [6,104]. Iron is a very useful magnetic adsorbent material but does not form a stable amalgam with mercury. However, a more stable amalgam can be obtained by doping the iron nanoparticles such as nZVI with metals that amalgamate with mercury, and this will ensure the effective formation of a more efficient and durable nanoadsorbent for $Hg^{2+}$ removal from water [7].

In addition, magnetite coated with *Lysinibacillus* sp. JLT12 was reported to efficiently reduce Cr(VI) to Cr(III) which is a less toxic and less soluble form of Cr [134]. The kinetics and spectroscopic data showed that the *Lysinibacillus* sp. JLT12 coating was able to remove the passive lepidocrocite and goethite layers of the magnetite to efficiently facilitate the Cr(VI) reduction process [134]. Also, Zhong et al. [80] through kinetics and spectroscopic studies attributed the formation of U-O and Mn-O-U bonds to the oxygen-containing groups in $\delta$-$MnO_2$@COF nanocomposite, which resulted in ultra-fast removal of $UO_2^{2+}$ radionuclide from water.

These extra abilities of inorganic nanoadsorbents demonstrate that efficient adsorbents can be designed and hold promise for efficient environmental remediation processes.

# 4. Environmental application

Hofacker et al. [135] in soil microcosm experiments demonstrated the potential application of inorganic nanoparticles in environmental remediation processes. The researchers set out to study mercury mobilization in flooded soil by incorporation into metallic copper and metal sulfide nanoparticles. The soil studied were topsoil samples obtained from the contaminated floodplain of the River Mulde in Germany, which were polluted with 0.91–1.26 mg $kg^{-1}$ Hg and 160–279 mg $kg^{-1}$ Cu. This level of pollution was 10 times the average content of European floodplain soils. Soil flooding experiments were carried out with air-dried soil (amended with lactate, which promotes activities of soil microorganisms including Fe and sulfate reducers) that was flooded with synthetic river water (containing 0.6 mM NaCl, 0.6 mM $CaSO_4$ and 0.3 mM $Mg(NO_3)_2$) and incubated at different periods and temperatures. Soil pore water samples were withdrawn, acidified, filtered and analysed for Hg, Cu, Fe, Mn and metal sulfides.

The study showed that the formation of Hg-Cu amalgam nanoparticles may be a common process induced by soil flooding under limited sulfate availability. However, during sulfate reduction, Hg was effectively incorporated into nanoparticulate metal sulfides. Figure 7 shows STEM-HAADF images

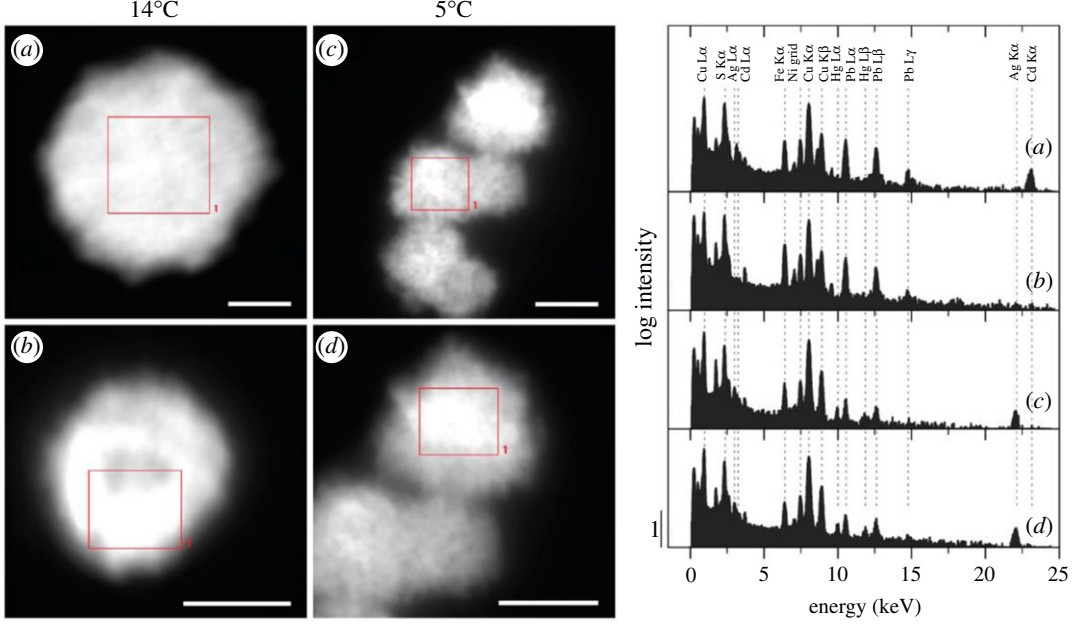

**Figure 7.** STEM-HAADF images and EDX spectra of metal sulfide nanoparticles formed after 12 days of flooding at 14°C (*a,b*) and after 33 days of flooding at 5°C (*c,d*). Scale bars represent 50 nm. EDX spectra were recorded on areas marked by red rectangles. Reprinted with permission from Hofacker *et al*. [135]. Copyright (2021) American Chemical Society.

and EDX spectra of metal sulfide nanoparticles formed after 12 days and 33 days of flooding at 14°C and 5°C, respectively. The tendency for different metals to precipitate with a limited amount of sulfide during soil flooding depends on the thermodynamic stability of the respective metal sulfides (Hg > Ag > Cu > Cd ∼ Pb > Zn > Fe). Also, coprecipitated Fe or Zn may be exchanged for Hg, Ag, Cu, Cd or Pb due to the lower stability of Fe and Zn monosulfides. Thus, based on the extremely high thermodynamic stability of HgS, higher soil Hg/Cu ratios are expected to favour the formation of pure HgS over mixed metal sulfide formation. The study showed that Hg incorporation into metallic Cu and metal sulfide nanoparticles can cause substantial colloidal Hg mobilization into the pore water. When exposed to oxygen, metal sulfide nanoparticles were found to be stable against oxidation in oxygenated water for up to several weeks. The formation of HgS was suggested to probably affect the extent of Hg methylation and volatilization which are the most toxic and dangerous aspects of Hg pollution.

Although Hofacker *et al*.'s [135] work demonstrates how soil floods induce the formation of inorganic nanoparticles depending on the soil's mineral composition, it does not show in real time how they could be applied to environmental remediation. Their work, however, suggests potential removal of Hg from aqueous solution using metal Cu and metal (Fe and Zn) sulfide nanoparticles through the formation of Hg-Cu amalgam and highly thermodynamic stable HgS nanoparticles precipitates, which can be filtered out of solution.

## 5. Conclusion

It has been established that inorganic nanoadsorbent materials address the problem of environmental pollution. Heavy metal contamination levels of over $100\,mg\,l^{-1}$ have been removed, attaining a very high adsorption capacity of less than or equal to $3449\,mg\,g^{-1}$ with inorganic nanoadsorbents with less than $2\,g\,l^{-1}$ adsorbent dosage. Although inorganic nanoadsorbents hold much promise in environmental remediation, there are very important bottlenecks that have to be addressed to harness their full potential. Some of the bottlenecks are:

(i) Metal nanoparticles such as Ag and Au are generally expensive for adsorption purposes. Thus, cheaper materials or earth-abundant metals would be more useful.

(ii) n-ZVI nanoparticles are very efficient but they are extremely reactive and may be unstable. More research efforts would be required to develop the core–shell counterparts to improve the stability.

(iii) Iron oxides nanoparticles have been widely used because they are magnetic and facilitate easy separation from solution; however, their heavy metal adsorption capacities are lower compared with CuO, ZnO, SnO and MgO nanoparticles which are not magnetic. Thus, doping nanoparticles with magnetic materials becomes an attractive strategy to improve the separability property of nanoadsorbents.

(iv) Metal sulfides nanoparticles have a higher affinity for heavy metals compared with metal oxides; however, they are less studied for remediation processes.

(v) It is worth noting that the control of selectivity in heavy metal adsorption has been associated with Pearson's theory of hard and soft acids and bases, but unfortunately, there is very little information to clearly explain the selectivity concept. The key responsible principle has been associated with the complexation of the heavy metals by functional groups on the inorganic nanoadsorbents surfaces. Although ligands attached to surfaces of nanoparticles improve the selectivity of nanoadsorbents towards target heavy metal, the process of nanoparticle surface functionalization can be very challenging and requires considerable research effort to improve on the qualities of inorganic nanoadsorbents.

In the design of good quality inorganic adsorbent, the nanoparticle synthesis method, selectivity, stability, regeneration and reusability, and adsorbent separation from solution are critical concerns. It is intended in the future that stable nanoadsorbents that are selective to specific pollutants, easy to synthesize (in an environmentally friendly manner) and separated from the solution, and also can perform other important functions are developed.

Ethics. This statement is not relevant to this work.

Data accessibility. Additional tables showing the BET surface area and point of zero charge of some inorganic nanoadsorbents have been provided in the electronic supplementary material.

Authors' contributions. M.B.M. designed, obtained literature and drafted the manuscript. D.J.L., N.O.B. and J.A.M.A. revised the manuscript critically for important intellectual content.

Competing interests. The authors declare no competing interests.

Funding. This work was funded by Leverhulme-Royal Society Africa Award-Postdoctoral Fellowship grant (grant no. LAF\R1\180018)

Acknowledgements. J.A.M.A., D.J.L. and M.B.M. wish to acknowledge the Leverhulme-Royal Society Africa Award-Postdoctoral Fellowship grant no. (LAF\R1\180018) for providing financial support for this project. The DFID-Royal Society Africa Capacity Building Initiative (ACBI) is acknowledged for support to J.A.M.A. and D.J.L. The Department of Chemistry, Kwame Nkrumah University of Science and Technology, Kumasi, is acknowledged for hosting the fellowship.

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
