## [Peer Review File · Royal Society Open Science]

Review History

RSOS-201485.R0 (Original submission)

Review form: Reviewer 1

Is the manuscript scientifically sound in its present form?

Yes

Are the interpretations and conclusions justified by the results?

Yes

Is the language acceptable?

Yes

Do you have any ethical concerns with this paper?

No

Have you any concerns about statistical analyses in this paper?

No

Recommendation?

Major revision is needed (please make suggestions in comments)

Comments to the Author(s)

In this review article, the authors summarized the recent works about the application of inorganic nanomaterials in the removal of heavy metal ions from aqueous solutions. The contents are helpful for readers to understand the recent works in this area. After reading the manuscript, I think it can be accepted for publication after revision.

Special comments:

1. Eqs. 1-3 are not clear. It is suggested to edit the equations to make them more clear.
2. Tables 2 and 3 are not necessary in this review. I suggest the authors to move them to Supporting Information.
3. In the Introduction section, several critical reviews should be added in the revised form such as: The Innovation, 2021, 2(1), 100076; Chemical Engineering Journal. 2021, 406, 127139; ACS ES&T Engineering. 2021, 1, 685-705.
4. In each section, the authors presented the results well. It is necessary to give the main results and conclusion for each section as this is easy for readers to understand the recent works, especially the advantages and disadvantages of different methods and materials.
5. If possible, it is better to add one section about the mechanism discussion at molecular level, such as spectroscopy analysis and theoretical calculations. The following papers should be added in the revised form such as: Chemosphere, 2021, 274, 129743; 2021, 262, 127901; Environmental Research, 2021, 196, 110349; Environmental Science: Nano, 2020, 7(11), 3303-3317; Science of the Total Environment. 2019, 685, 986-996; Environmental Science & Technology. 2019, 53, 6454-6461.
6. In the Conclusion section, it is better to give the main challenges in future about the application of inorganic nanomaterials in environmental pollution management.

Review form: Reviewer 2

Is the manuscript scientifically sound in its present form?

Yes

Are the interpretations and conclusions justified by the results?

Yes

Is the language acceptable?

Yes

Do you have any ethical concerns with this paper?

No

Have you any concerns about statistical analyses in this paper?

No

Recommendation?

Major revision is needed (please make suggestions in comments)

Comments to the Author(s)

The paper by Mensah et al. summarizes the fabrication and utilization of different inorganic nanomaterials for heavy metal pollution and other environmental remediation. The inorganic nanomaterials discussed include metal, metal oxide, and metal sulfide, etc. The important strategies for improving the sensitivity, selectivity, and reusability of the inorganic nanomaterials for environmental remediation have also been highlighted. Overall, the topic of the review is

interesting and appealing to a broad audience. The content is relatively easy to follow. However, some revisions are necessary before this review can be accepted as detailed below:

1. Please include some figures showing the morphology and performance of the different inorganic nanoadsorbents (as case studies) in heavy metal pollution remediation and other environmental remediation. While Tables are always good, it is better to include some case studies involving different nanoadsorbents.
2. For the synthesis part, it is better to include some schematic illustrations to provide visual aids for the readers.
3. One Table summarizing the advantages and drawbacks of the different synthetic methods can be given.
4. The advantages and disadvantages of different inorganic adsorbents can also be summarized in one Table.
5. Some comments and suggestions for future research directions on the development of nanoadsorbents for environmental remediation can be provided in the Conclusions section.
6. Other relevant reviews on the development of nanomaterials for environmental remediations, such as *New J. Chem.*, 43, 15846-15856 (2019); *Molecules*, 23(7), 1760 (2018); *Chemical Engineering Journal*, 408, 127991 (2021); *Environmental Nanotechnology, Monitoring & Management*, 13, 100279 (2020), and *Small*, 3, 1800512 (2019) can be mentioned and cited in the Introduction.

Decision letter (RSOS-201485.R0)

Dear Dr Mensah:

Title: Heavy metal pollution and the role of inorganic nanomaterials in environmental remediation

Manuscript ID: RSOS-201485

The editor assigned to your manuscript has now received comments from reviewers. We would like you to revise your paper in accordance with the referee and Subject Editor suggestions which can be found below (not including confidential reports to the Editor). Please note this decision does not guarantee eventual acceptance.

Please submit your revised paper before 28-Jul-2021. Please note that the revision deadline will expire at 00.00am on this date. If we do not hear from you within this time then it will be assumed that the paper has been withdrawn. In exceptional circumstances, extensions may be possible if agreed with the Editorial Office in advance. We do not allow multiple rounds of revision so we urge you to make every effort to fully address all of the comments at this stage. If deemed necessary by the Editors, your manuscript will be sent back to one or more of the original reviewers for assessment. If the original reviewers are not available we may invite new reviewers.

To revise your manuscript, log into <http://mc.manuscriptcentral.com/rsos> and enter your Author Centre, where you will find your manuscript title listed under "Manuscripts with Decisions." Under "Actions," click on "Create a Revision." Your manuscript number has been

appended to denote a revision. Revise your manuscript and upload a new version through your Author Centre.

On behalf of the Subject Editor Professor Anthony Stace and the Associate Editor Dr Nadia Martinez Villegas.

RSC Associate Editor:

Comments to the Author:

Thank you for considering RSOS for your manuscript submission. Your manuscript has been carefully examined and, in view of the criticisms of the reviewers, major revision is needed. We look forward to seeing the revised version.

RSC Subject Editor:

Comments to the Author:

(There are no comments.)

Reviewers' Comments to Author:

Reviewer: 1

Comments to the Author(s)

In this review article, the authors summarized the recent works about the application of inorganic nanomaterials in the removal of heavy metal ions from aqueous solutions. The contents are helpful for readers to understand the recent works in this area. After reading the manuscript, I think it can be accepted for publication after revision.

Special comments:

1. Eqs. 1-3 are not clear. It is suggested to edit the equations to make them more clear.
2. Tables 2 and 3 are not necessary in this review. I suggest the authors to move them to Supporting Information.

3. In the Introduction section, several critical reviews should be added in the revised form such as: *The Innovation*, 2021, 2(1), 100076; *Chemical Engineering Journal*, 2021, 406, 127139; *ACS ES&T Engineering*, 2021, 1, 685-705.
4. In each section, the authors presented the results well. It is necessary to give the main results and conclusion for each section as this is easy for readers to understand the recent works, especially the advantages and disadvantages of different methods and materials.
5. If possible, it is better to add one section about the mechanism discussion at molecular level, such as spectroscopy analysis and theoretical calculations. The following papers should be added in the revised form such as: *Chemosphere*, 2021, 274, 129743; 2021, 262, 127901; *Environmental Research*, 2021, 196, 110349; *Environmental Science: Nano*, 2020, 7(11), 3303-3317; *Science of the Total Environment*, 2019, 685, 986-996; *Environmental Science & Technology*, 2019, 53, 6454-6461.
6. In the Conclusion section, it is better to give the main challenges in future about the application of inorganic nanomaterials in environmental pollution management.

Reviewer: 2

Comments to the Author(s)

The paper by Mensah et al. summarizes the fabrication and utilization of different inorganic nanomaterials for heavy metal pollution and other environmental remediation. The inorganic nanomaterials discussed include metal, metal oxide, and metal sulfide, etc. The important strategies for improving the sensitivity, selectivity, and reusability of the inorganic nanomaterials for environmental remediation have also been highlighted. Overall, the topic of the review is interesting and appealing to a broad audience. The content is relatively easy to follow. However, some revisions are necessary before this review can be accepted as detailed below:

1. Please include some figures showing the morphology and performance of the different inorganic nanoadsorbents (as case studies) in heavy metal pollution remediation and other environmental remediation. While Tables are always good, it is better to include some case studies involving different nanoadsorbents.
2. For the synthesis part, it is better to include some schematic illustrations to provide visual aids for the readers.
3. One Table summarizing the advantages and drawbacks of the different synthetic methods can be given.
4. The advantages and disadvantages of different inorganic adsorbents can also be summarized in one Table.
5. Some comments and suggestions for future research directions on the development of nanoadsorbents for environmental remediation can be provided in the Conclusions section.
6. Other relevant reviews on the development of nanomaterials for environmental remediations, such as *New J. Chem.*, 43, 15846-15856 (2019); *Molecules*, 23(7), 1760 (2018); *Chemical Engineering Journal*, 408, 127991 (2021); *Environmental Nanotechnology, Monitoring & Management*, 13, 100279 (2020), and *Small*, 3, 1800512 (2019) can be mentioned and cited in the Introduction.

Author's Response to Decision Letter for (RSOS-201485.R0)

See Appendix A.

RSOS-201485.R1 (Revision)

Review form: Reviewer 1

Is the manuscript scientifically sound in its present form?

Yes

Are the interpretations and conclusions justified by the results?

Yes

Is the language acceptable?

Yes

Do you have any ethical concerns with this paper?

No

Have you any concerns about statistical analyses in this paper?

No

Recommendation?

Accept as is

Comments to the Author(s)

The authors revised the manuscript carefully. I recommend for publication.

Review form: Reviewer 2

Is the manuscript scientifically sound in its present form?

Yes

Are the interpretations and conclusions justified by the results?

Yes

Is the language acceptable?

Yes

Do you have any ethical concerns with this paper?

No

Have you any concerns about statistical analyses in this paper?

No

Recommendation?

Accept as is

Comments to the Author(s)

The authors have addressed my previous comments thoroughly and added additional information necessary to improve the quality of this paper. Therefore, I am happy to accept the manuscript in the present form.

Decision letter (RSOS-201485.R1)

Dear Dr Mensah:

Title: Heavy metal pollution and the role of inorganic nanomaterials in environmental remediation

Manuscript ID: RSOS-201485.R1

It is a pleasure to accept your manuscript in its current form for publication in Royal Society Open Science. The chemistry content of Royal Society Open Science is published in collaboration with the Royal Society of Chemistry.

Yours sincerely,
Dr Ellis Wilde
Publishing Editor, Journals

On behalf of the Subject Editor Professor Anthony Stace and the Associate Editor Dr Nadia Martinez Villegas.

RSC Associate Editor
Comments to the Author:
This paper has been carefully revised and can be accepted now.

RSC Subject Editor
Comments to the Author:
(There are no comments.)

Reviewer(s)' Comments to Author:

Reviewer: 1

Comments to the Author(s)

The authors revised the manuscript carefully. I recommend for publication.

Reviewer: 2

Comments to the Author(s)

The authors have addressed my previous comments thoroughly and added additional information necessary to improve the quality of this paper. Therefore, I am happy to accept the manuscript in the present form.

Appendix A

Response to Reviewers' Comments

Reviewer: 1

Comment

In this review article, the authors summarized the recent works about the application of inorganic nanomaterials in the removal of heavy metal ions from aqueous solutions. The contents are helpful for readers to understand the recent works in this area. After reading the manuscript, I think it can be accepted for publication after revision.

Response: We thank the reviewer for this encouraging comment.

Special comments:

1. Eqs. 1-3 are not clear. It is suggested to edit the equations to make them more clear.

Response: Eqs. 1-3 have been redrawn and made clearer than before

2. Tables 2 and 3 are not necessary in this review. I suggest the authors to move them to Supporting Information.

Response: Tables 2 and 3 have been moved to the Supplementary Information

3. In the Introduction section, several critical reviews should be added in the revised form such as: The Innovation, 2021, 2(1), 100076; Chemical Engineering Journal. 2021, 406, 127139; ACS ES&T Engineering. 2021, 1, 685-705.

Response: Some of these references suggested have been added to the revised manuscript.

4. In each section, the authors presented the results well. It is necessary to give the main results and conclusion for each section as this is easy for readers to understand the recent works, especially the advantages and disadvantages of different methods and materials.

Response: Yes, we agree. To make it easy for readers, Tables summarizing the advantages and disadvantages of the different methods and nanomaterials have been added (Tables 2 and 3) to the revised manuscript. Conclusions for each major sections have been added and highlighted in the revised manuscript.

5. If possible, it is better to add one section about the mechanism discussion at molecular level, such as spectroscopy analysis and theoretical calculations. The following papers should be added in the revised form such as: Chemosphere, 2021, 274, 129743; 2021, 262, 127901; Environmental Research, 2021, 196, 110349; Environmental Science: Nano, 2020, 7(11), 3303-3317; Science of the Total Environment. 2019, 685, 986-996; Environmental Science & Technology. 2019, 53, 6454-6461.

Response: A brief statement on theoretical calculations has been added (and highlighted) in the revised manuscript. A whole section on some theoretical calculations was not added because at the

present, there are enough data on the adsorption of heavy metals by inorganic nanomaterials which show much promise. However, theoretical calculations on mechanisms at molecular level would be considered in future publications. Some spectroscopic analyses have been added. Some of the suggested articles have been added and those areas have been highlighted in the revised manuscript. We thank the reviewer for this good suggestion to improve our paper.

6. In the Conclusion section, it is better to give the main challenges in future about the application of inorganic nanomaterials in environmental pollution management.

Response: The conclusion section has been rewritten to give the main challenges and the future outlook for application of nanomaterials in environmental pollution management.

Reviewer: 2

Comments to the Author(s)

The paper by Mensah et al. summarizes the fabrication and utilization of different inorganic nanomaterials for heavy metal pollution and other environmental remediation. The inorganic nanomaterials discussed include metal, metal oxide, and metal sulfide, etc. The important strategies for improving the sensitivity, selectivity, and reusability of the inorganic nanomaterials for environmental remediation have also been highlighted. Overall, the topic of the review is interesting and appealing to a broad audience.

Response: We thank the reviewer for this encouraging comment.

The content is relatively easy to follow. However, some revisions are necessary before this review can be accepted as detailed below:

1. Please include some figures showing the morphology and performance of the different inorganic nanoadsorbents (as case studies) in heavy metal pollution remediation and other environmental remediation. While Tables are always good, it is better to include some case studies involving different nanoadsorbents.

Response: We have added a number of new figures that are complementary to the case studies that are in the text, (Figure 1, Figure 2, Figure 7), that show both the morphology of the materials (from electron microscopy) and the performance of the material in remediation applications. We thank the reviewer for this good suggestion to improve our paper.

2. For the synthesis part, it is better to include some schematic illustrations to provide visual aids for the readers.

Response: A schematic diagram has been added to provide a visual representation to readers.

3. One Table summarizing the advantages and drawbacks of the different synthetic methods can be given.

Response: A table summarizing the advantages and disadvantages of the different synthetic methods for nanomaterials have been added as Table 3 in the revised manuscript.

4. The advantages and disadvantages of different inorganic adsorbents can also be summarized in one Table.

Response: A table summarizing the advantages and disadvantages of the different inorganic nanomaterials have been added as Table 2 in the revised manuscript.

5. Some comments and suggestions for future research directions on the development of nanoadsorbents for environmental remediation can be provided in the Conclusions section.

Response: The conclusion section has been rewritten to give the main challenges and the future outlook for application of nanomaterials in environmental pollution management.

6. Other relevant reviews on the development of nanomaterials for environmental remediations, such as *New J. Chem.*, 43, 15846-15856 (2019); *Molecules*, 23(7), 1760 (2018); *Chemical Engineering Journal*, 408, 127991 (2021); *Environmental Nanotechnology, Monitoring & Management*, 13, 100279 (2020), and *Small*, 3, 1800512 (2019) can be mentioned and cited in the Introduction.

Response: Some of the suggested articles have been added and those areas have been highlighted in the revised manuscript.